# Effect of cesarean section on initiation of breast feeding: Findings from 2016 Ethiopian Demographic and Health Survey

**Getnet Gedefaw**[1]☯*, **Martha H. Goedert**[2‡], **Eskeziaw Abebe**[1‡], **Asmamaw Demis**[3‡]

**1** Department of Midwifery, College of Health Sciences, Woldia University, Woldia, Ethiopia, **2** Department of Health Promotions, College of Public Health, University of Nebraska Medical Center, Omaha, Nebraska, United States of America, **3** Department of Nursing, College of Health Sciences, Woldia University, Woldia, Ethiopia

☯ These authors contributed equally to this work.
‡ These authors also contributed equally to this work.
* gedefawget@gmail.com

**Data Availability Statement:** All the datasets we used for this study are publicly available from the DHS Program website http://dhsprogram.com/data.

## Abstract

### Background

Early initiation of breast feeding has great importance for both mothers and newborns. Despite, recommendations for exclusive and early initiation of breast feeding within one hour of birth, Ethiopia reports that 58% of infants under six months of age are exclusively breastfed. Cesarean deliveries may affect timing of breastfeeding initiation, establishment of milk supply and infant breastfeeding interest compared to vaginal deliveries. The aim of this study was to assess the impact of cesarean delivery on breastfeeding initiation.

### Methods

A cross-sectional study was conducted with a total of 7115 study participants from the 2016 Ethiopian Demographic Health Survey (EDHS). Both descriptive and analytical statistical analysis was employed. Univariable and multivariable logistic regression analyses were used to identify any associations between variables. Odds ratios with its corresponding 95% confidence intervals (CI) were reported. During multivariable analysis, variables with $p$-value < 0.05 were considered as statistically significant.

### Results

The prevalence of late initiation of breast feeding among women with their last live birth was 25.03% (95%CI; 20.5–32.2). Significant factors associated with late initiation of breastfeeding were cesarean delivery [AOR = 4.06 (95%CI, 2.66–6.2)], primipara mother [AOR = 1.45 (95%CI, 1.13–1.7)], and having an unplanned pregnancy [AOR = 1.35(95%CI, 1.1–1.65)]. Positively associated with early initiation of breastfeeding was the mother's age, for women between 20–34 years-old. This association, reported as a negative association of late initiation of breast feeding was [AOR = 0.77(95%CI, 0.61–0.98)].

**Funding:** No. There is no specific funding for this study.

**Competing interests:** The authors have declared that no competing interests exist.

**Abbreviations:** CI, Confidence Interval; IBF, Initiate Breast feeding; OR, Odd Ratio; UNICEF, United nation International Children's Emergency Fund; WHO, World Health Organization.

## Conclusion

Cesarean delivery adversely affects the initiation of breast feeding. Cesarean delivery, unplanned pregnancy, and being primiparous, were associated with late initiation of breast-feeding. Women between the ages of 20–34 years of age had a reduced chance of late initiation of breastfeeding. Providing counseling regarding the strategy and importance of early initiation of breast feeding, could have crucial importance for the mother and her newborn.

## Background

Breast feeding is essential for optimal growth, development and health of neonates and infants. According to the World Health Organization (WHO) and the United Nation'sInternational Children's Emergency Fund (UNICEF), breastfeeding should be initiated within one hour afterdelivery. In addition to promoting early initiation of breastfeeding for all neonates, both organizations recommend sustaining exclusive breastfeeding through six months of age [1].

Early initiation of breastfeeding increases the chances of a successful continuation of breast-feeding. Breastfeeding is associated with reduced infant and under-5 mortality and morbidity, protects the neonate from infection and promotes ideal nutrition with lower acute and severe malnutrition [2].

Ethiopia's Federal Ministry of Health (FMoH) established the National Nutrition Program II (NNP II) along with the National Guideline on Adolescent, Maternal, Infant, and Young Child Nutrition (AMIYCN)to promote optimal feeding and care practices that follow international recommendations (FDRE, 2106b). After this FMOH initiative research reported that less than three-quarter (73%) were breastfed within 1 hour of birth [3].

Demographic health surveys from 2000 to 2013, across low and middle income countries (LMIC) in Asia, Latin America, the Middle East, Europe, and Sub-Saharan Africa reported that the average rate of early initiation of breastfeeding was 39% and avoidance of pre-lacteal feeding was 49.2% [4].

The research reports that associated with cesarean deliveries are adverse maternal health outcomes including a high incidence of puerperal sepsis, incisional pain, obstetric hemorrhage and bladder and bowel complaints. Adverse fetal outcomes associated with surgical birth include hypoglycemia, respiratory distress syndrome, jaundice, and decreased success with breastfeeding, including increased early cessation of lactation [5–8].

Research supports the impact of delays in breastfeeding initiation as associated with cesar-eandelivery. Delays in breastfeeding initiation along with feeds later than one hour after birth further could compromise successful lactation. Separating mothers from their newborns; also impacts the establishment of an adequate milk supply by practices linked to decreased time at the breast, decreased neonatal interest in feeding at the breast, insufficient milk production and ultimately shortened breastfeeding duration [6, 9, 10].

Factors identified to impact early onset of breastfeeding in Ethiopia included mode of delivery, maternal education, delivery site, gestational age, type of delivery attendant, type of prenatal attendant, prenatal guidance on breastfeeding, postpartum counseling about breastfeeding and [11–13].

According to the systematic and meta-analysis conducted in china reported that the practice of maternal-infant skin to skin contact after birth has been increasingly considered as an efficient way to promote breastfeeding initiation However, the feasibility of an intervention of

skin to skin contact after caesarean delivery in the operating room to improve breastfeeding as well as maternal satisfaction is challenging [14].

Cesarean delivery has a negative impact on early breastfeeding might be mediated through processes that delay the onset of lactation, disrupt mother-infant interaction, or inhibit infant suckling. Timing of the first feeding, postoperative care routines after cesarean delivery, interrupt bonding, delay mothers holding their infants, are the potential mechanisms of early breastfeeding reduction [15].

Despite, different strategy has been implemented; the prevalence of exclusive and early initiation of breastfeeding is underscored, accounted for 59% and 75.7% respectively showed that the level of exclusive and early initiation of breastfeeding is stagnant over years in low and middle income countries particularly in Ethiopia [16, 17].

In light of the research and WHO recommendation that neonatal outcomes are improved by early and exclusive breastfeeding, this gap in Ethiopian maternal child health practice is concerning. Globally, the research points to breastfeeding as important to enhance bonding provide IGA, microbiome and immunoglobulin protections against infections for the neonate and infant. As early initiation of breastfeeding is related to successful exclusive breastfeeding for 6 months, this research is an important step to increasing Ethiopian practices related to lactation for maternal and child health. Furthermore, this research aims to determine if cesarean delivery may be a factor impacting early and exclusive breastfeeding in Ethiopia to overlook the impact of cesarean section deliveries on timely initiation of breastfeeding practice in Ethiopia using 2016 Ethiopian Demographic Health Survey Data [3].

## Methods and materials

### Study area and design

The study was conducted in Ethiopia, in the East Horn of Africa. This research analyzed factors reported by the 2016 Ethiopia Demographic and Health Survey (EDHS) from the fourth version of the survey [3]. The 2016 EDHS data a community based cross-sectional population-based survey data implemented by the Central Statistical Agency (CSA) collected from January 18, 2016, to June 27, 2016.

### Data and sampling procedures

Data for this study were retrieved from the 2016 EDHS, which used a weighted multistage, stratified cluster sampling approach. The 2016 EDHS data employed a two-stage stratified cluster sampling procedure for nine regional states and two city administrations. Initially, each region was stratified into urban and rural areas, yielding 21 sampling strata. After stratification, a total of 645 Enumeration areas(202 in urban areas and 443 in rural areas) were selected using a probability proportional to the Enumeration size. Enumeration size was based on the 2007 Ethiopia population and housing census and with independent selection in each sampling stratum. The number of households served as a sampling frame for the selection of households. The numbers of households per cluster were fixed and selected with an equal probability systematic selection from the newly created household listing. Lastly, all women age 15–49 who were either permanent residents of the selected households or visitors who stayed in the household the night before the survey were eligible to be interviewed for interviewer-administered structured questionnaires. A detailed description study design and methods of data collection for the 2016 EDHS are available elsewhere (3). A total of 15,683 women aged 15–49 years were interviewed in the 2016 EDHS, of which 7,590 women had at least one live birth in the last 5 years prior to the survey. Four hundred seventy five women were

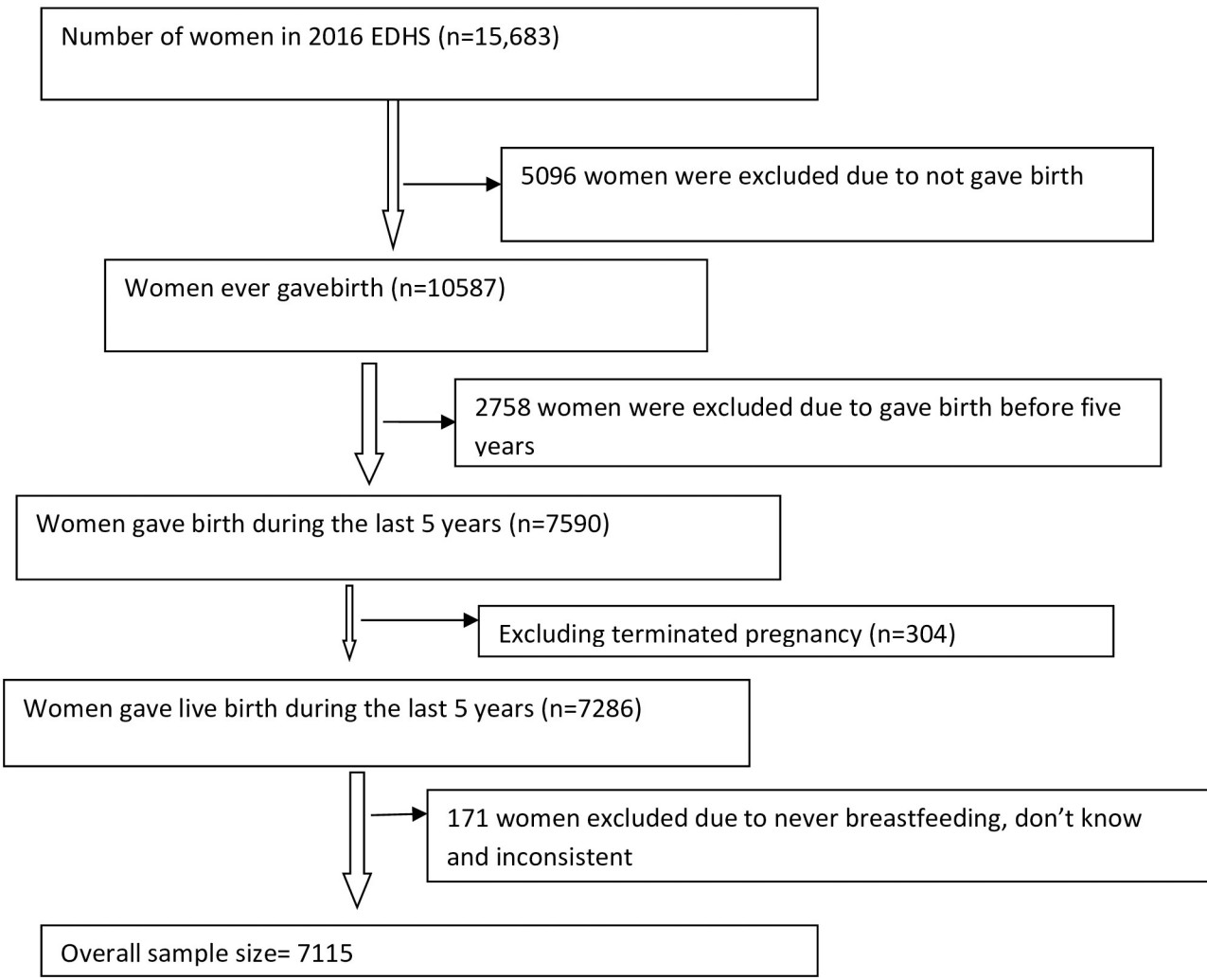

**Fig 1. Schematic presentation showing sampling procedure in the 2016 Ethiopia Demographic and Health Surveys (EDHS); Ethiopia.**

excluded from the sample for reasons related to termination or no breastfeeding history. Those women who were excluded were women who had terminated their last pregnancy (n = 304) or who gave a history that was inconsistent with breastfeeding (n = 171 ('do not know', 'inconsistent' or 'never breastfed')). The total number of women included for this study was 7,115 women (Fig 1).

## Measurement of outcomes

**Initiation of breast feeding.** Initiation of breastfeeding among women who recently gave birth was the primary outcome of the study which coded as early"0" and late"1" initiation of breastfeeding.

**Exposure measurement.** *Mode of delivery*. Mode of delivery wasdefined and categorizedas either "vaginal delivery" or "cesarean delivery".

*Covariates*. Socio-demographic, obstetrics, newborn and reproductive health related characteristics were included as covariates.

*Socio-demographic characteristics included*. Residence, age of the mother, educational status of the mother, educational status of her husband, religion, marital status, occupation of the mother, body mass index, and wealth index.

*Obstetrics and reproductive health characteristics included*. Parity, status of pregnancy, antenatal care, prenatal attendant, delivery attendant, mode of delivery, place of delivery, gravidity, number of antenatal care visit, counseling on breastfeeding, and number of children.

*Neonatal characteristics*. included age of the newborn, sex, birth weight and gestational age.

**Data processing and analysis.** Data was analyzed using SPSS version 24 statistical software. Complex sample survey (stratified/clustered) sampling designs was used to correctly calculateunequal probabilities of selection with weighted data. Rao-Scott chi-square was used to examine the univariable associations between each covariate and breastfeeding initiation via adjusting complex survey sampling. Both univariable and multivariable logistic regression analyses were performed to identify the impact of cesarean delivery on initiation of breast feeding.

Descriptive statistics such as frequencies mean and proportions were calculated. Odds ratios with its corresponding 95% confidence intervals (CI) were reported. Factors in the univariable logistic regression with a *p*-value of less than 0.25 [18] were a candidate to be fitted into the multivariable logistic regression to control the possible effects of confounders and asses association between cesarean section and initiation of breast feeding. In the multivariable analysis, variables with a *p*-value $< 0.05$ were declared as statistically significant.

**Operational definition.** *Timely initiation of breastfeeding*. Refers to if a mother who put her baby to breast within one hour following delivery [1].

*Cesarean section*. A surgical procedure involving incision of the walls of the abdomen and uterus for delivery of offspring.

**Ethical approval and consent to participate.** All the available datasets were obtained from the DHS website (https://dhsprogram.com/) through registering with the DHS website no ethical approval was required. Institutional Review Board of Woldia University waived the requirement for informed consent for EDHS data. All the authors have no special access privileges to this dataset. To access and get the data, authors should register and log in. While, we requested the title of the project, co-authors name, email addresses and brief description about the study should be clearly stated. Then after, the researchers continue to select the country, dataset and the year of the survey. Within a few days after requested, the demographic and health survey team will get permission to download the dataset via email of the corresponding author. After permission was stated, the author can log in and select the specific data with the important format what the author wants.

## Results

### Socio-demographic characteristics

In this study more than three-quarter 6214(87.34%) of the study participants were living in rural area. Regarding the age of the women, most of the study participants 5042(70.86%) was under the age group of 20–34. Out of the total study participants, more than half of them 5738 (80.65%) were exposed to mass media (Table 1).

### Obstetrics, newborn and reproductive health characteristics

Out of 7115 women, more than 4509(63.37%) of the study participants had antenatal care follow-up. More than three-quarter of the study participants 5758(80.93%) and 5817(81.76%) were multipara and multigravida women respectively. Regarding place of delivery, more than

**Table 1. Socio-demographic characteristics.**

| Variables | Frequency | Percent (%) |
|---|---|---|
| **Educational level of the women** | | |
| No education | 4488 | 63.08 |
| Primary education | 2024 | 28.45 |
| Secondary education | 390 | 5.48 |
| Higher education | 213 | 2.99 |
| **Residence** | | |
| Urban | 901 | 12.66 |
| Rural | 6214 | 87.34 |
| **Marital status** | | |
| Currently unmarried | 429 | 6.03 |
| Currently married | 6686 | 93.97 |
| **Religion** | | |
| Orthodox | 2732 | 38.40 |
| Muslim | 2610 | 36.68 |
| Other(Protestant, Catholic, Traditional) | 1773 | 24.92 |
| **Maternal age** | | |
| ≤19 | 1043 | 14.66 |
| 20–34 | 5042 | 70.86 |
| ≥35 | 1030 | 14.48 |
| **Occupation** | | |
| Working | 2025 | 28.46 |
| Not working | 5090 | 71.54 |
| **Wealth index** | | |
| Poorer | 1570 | 22.07 |
| Poorest | 1542 | 21.67 |
| Middle | 1490 | 20.94 |
| Richer | 1331 | 18.71 |
| Richest | 1182 | 16.61 |
| **Educational level of the husband** | | |
| No education | 3189 | 47.69 |
| Primary | 2574 | 38.50 |
| Secondary and above | 923 | 13.81 |
| **Exposed to media** | | |
| Yes | 1377 | 19.35 |
| No | 5738 | 80.65 |

half of the women 6313(88.73%) gave birth at a health facility. More than half of the women 3820(53.69%) were assisted during birth by a skilled health professional (Table 2).

## Prevalence of late initiation of breastfeeding

This study revealed that the prevalence of late initiation of breastfeeding among women who gave live birth in the last five years preceding the survey was 25.03% (95%CI; 20.5–32.2) whereas 61.2% (95%CI; 51.8–73.2) of the women experienced delayed or late initiation of breastfeeding in Ethiopia.

**Table 2. Obstetrics, newborn and reproductive health characteristics.**

| Antenatal care booking | Frequency | Percent (%) |
|---|---|---|
| Yes | 4509 | 63.37 |
| No | 2606 | 36.63 |
| **Parity** | | |
| Multipara | 5758 | 80.93 |
| Primipara | 1357 | 19.07 |
| **Gravidity** | | |
| Multigravida | 5817 | 81.76 |
| Primigravida | 1298 | 18.24 |
| **Age at first birth** | | |
| ≤19 | 4481 | 62.98 |
| ≥20 | 2634 | 37.02 |
| **Age at first marriage** | | |
| ≤18 | 5129 | 72.63 |
| ≥19 | 1933 | 27.37 |
| **Status of pregnancy** | | |
| Planned | 5258 | 73.9 |
| Not planned | 1857 | 26.1 |
| **Mode of delivery** | | |
| Cesarean section | 165 | 2.3 |
| Vaginal delivery | 6950 | 97.7 |
| **Sex of the newborn** | | |
| Male | 3652 | 51.3 |
| Female | 3463 | 48.7 |
| **Place of delivery** | | |
| Health facility | 6313 | 88.73 |
| Home | 802 | 11.27 |
| **Distance from health facility** | | |
| Big problem | 4151 | 58.34 |
| Not big problem | 2964 | 41.66 |
| **Breastfeeding counseling during postnatal period** | | |
| Yes | 1813 | 25.48 |
| No | 5302 | 74.52 |
| **Health provider counseling on newborn danger signs** | | |
| Yes | 789 | 11.09 |
| No | 6326 | 88.91 |
| **Prenatal attendant** | | |
| Skilled | 4486 | 63.05 |
| Non skilled | 2629 | 36.95 |
| **Delivery attendant** | | |
| Skilled | 3820 | 53.69 |
| Non-skilled | 3295 | 46.31 |
| **Gestational age at birth** | | |
| Large for gestational age | 2234 | 31.4 |
| Average for gestational age | 2910 | 40.9 |
| Small for gestational age | 1915 | 26.9 |
| Unknown | 56 | 0.8 |
| **Number of ever children** | | |
| 1–2 | 2471 | 34.73 |
| 3–5 | 2683 | 37.71 |

*(Continued)*

Table 2. (Continued)

| Antenatal care booking | Frequency | Percent (%) |
|---|---|---|
| 6 and more | 1961 | 27.56 |
| **Number of antenatal care visit** | | |
| No | 2606 | 36.6 |
| 1–2 | 896 | 12.6 |
| 3 and more | 3613 | 50.8 |
| **Body mass index** | | |
| Underweight | 1370 | 19.25 |
| Normal | 5089 | 71.53 |
| Over weight | 464 | 6.52 |
| Obese | 192 | 2.7 |

## Association between cesarean section and initiation of breastfeeding

In this study, there is a negative association between elective cesarean delivery and early breast-feeding. Womenwho were delivered by cesarean delivery were four times more likely initiate breast feeding late as compared with women who gave birth through vaginal delivery (AOR:4.06, 95%CI:2.66–6.2). This might be due to that cesarean delivery is carried out for a variety of reasons, including maternal illness and fetal compromise, which may result reduce breastfeeding success (Table 3).

## Discussion

Cesarean deliveryis increasing from 2000 to 2016 according to Ethiopian Demographic Health Survey, from time trend analysis. Two percent of births were by cesarean delivery in 2016 compared to one percent of cesarean deliveries in 2011 (EDHS, 2011). Access to cesarean section can reduce different maternal and neonatal complications; however having a delivery by cesarean section is associated with late initiation or delayed initiation of breastfeedingas well as with discontinuation of exclusive breastfeeding.

This study revealed that the result of late initiation of breastfeeding among women who gave the last live birth was 25.03% (95%CI; 20.5–32.2). In this population level study, the prevalence of delayed or late initiation of breastfeeding among women who delivered by cesarean section was 61.2% (95%CI; 51.8–73.2).

The finding of this study is in line with the study conducted in Ethiopia [66.2%], and Sudan [52%]. The finding of this study is lower than the study conducted in Bahir Dar town (87.0%), Australia 98%, and Saudi Arabia (77.8%). This difference may be due to health policy difference among the countries and due to the difference in sociodemographic characteristics. In addition, the discrepancy of the study might be due to this study analyzed using huge data at national level, resulted accuracy and representative data as compared to the single studies. Besides, in this population, only 2.3% of women delivered by caesarean section as compared to vaginal delivery, accounted 97.7%, and this makes the effect of cesarean section on delayed initiation of breast feeding might be small [19–23].

The odds of having delayed initiation of breastfeeding is four times higher among women who had cesarean deliveries [AOR = **4.06(95%CI; 2.66–6.2)**] as compared to women who delivered vaginally. This study finding was supported by the study done in Uganda, Turkey, Ethiopia, south Sudan, Brazil, India, and Nigeria. Literatures supported mode of delivery is the major determinant factor for initiation of breastfeeding, this might be due cesarean section affects the well-being and psychology of the women associated with major abdominal surgery,

**Table 3. Univariable and multivariable logistic regression.**

| Variable | Initiation of breast feeding | | COR | AOR | p-value |
|---|---|---|---|---|---|
| | Early | Late | | | |
| **Mode of delivery** | | | | | |
| Vaginal delivery | 5270(74.1%) | 1680(23.6%) | 1.0 | **1.0** | |
| Cesarean section | 64(0.9%) | 101(1.4%) | 4.97(3.27–7.55) | **4.06(2.66–6.2)** | **0.000** |
| **Maternal age** | | | | | |
| ≤19 | 750(10.5%) | 293(4.1%) | 1.0 | **1.0** | |
| 20–34 | 3851(54.1%) | 1191(16.7%) | 0.76(0.6–0.97) | **0.77(0.61–0.98)** | **0.036** |
| ≥35 | 734(10.3%) | 296(4.16%) | 0.97(0.72–1.31) | 0.83(0.57–1.17) | 0.26 |
| **Status of pregnancy** | | | | | |
| Planned | 4010(56.4%) | 1248(7.5%) | 1.0 | **1.0** | |
| Unplanned | 1323(18.6%) | 334(17.5%) | 1.29(1.05–1.59) | **1.35(1.1–1.65)** | **0.002** |
| **Health provider counseling on newborn danger signs** | | | | | |
| Yes | 535(7.5%) | 254(3.6%) | 1.0 | **1.0** | |
| No | 4587(64.5%) | 1739(24.4%) | 0.8(0.64–0.99) | 0.93(0.75–1.17) | 0.59 |
| **Parity** | | | | | |
| Primipara | 935(13.2%) | 422(5.9%) | 1.45(1.18–1.78) | **1.45(1.13–1.7)** | **0.004** |
| Multipara | 4397(61.8%) | 1361(19.1%) | 1.0 | **1.0** | |
| **Delivery attendant** | | | | | |
| Skilled | 2823(39.7%) | 997(14%) | 1.0 | **1.0** | |
| Non-skilled | 2299(32.3%) | 996(14%) | 1.23(1.03–1.47) | 1.06(0.87–1.29) | 0.55 |
| **Prenatal attendant** | | | | | |
| Skilled | 3167(44.5%) | 1319(18.5%) | 1.0 | **1.0** | |
| Non-skilled | 1955(27.5%) | 674(9.5%) | 0.83(0.68–1.01) | 0.87(0.69–1.08) | 0.21 |
| **Educational level of the women** | | | | | |
| No education | 3258(45.8%) | 1230(17.3%) | 0.55(0.35–0.88) | 0.93(0.55–1.58) | 0.79 |
| Primary education | 1445(20.3%) | 579(8.1%) | 0.59(0.37–0.94) | 0.84(0.49–1.42) | 0.5 |
| Secondary education | 292(4.1%) | 98(1.4%) | 0.49(0.28–0.85) | 0.59(0.33–1.08) | 0.85 |
| Higher education | 127(1.8%) | 86(1.2%) | 1.0 | **1.0** | |

NB: 1.0 = Reference.

and post-surgical procedures for the women and her newborn routine procedures may delay the initiation of breastfeeding due to the physiology of lactation during the early postpartum period. Besides, cesarean section usually stays under various obstetric related health problems, effect of general anesthesia, pain and tiredness [24–33].

Primipara women were [**AOR = 1.45(1.13–1.7)**] 1.45 times more likely to initiate breastfeeding lately than multipara women. The result of this study is consistent and supported by the study employed in India and from eight country analysis. The possible justification might be due to intensity of mother's perception of breast engorgement from one to three days after birth was significantly more pronounced in multiparous women as compared to primiparous women [29, 30]. This might be due to that primipara women are at risk of obstructed labor due to pelvic inadequacy resulting cesarean section as a result the likelihood of early initiation of breastfeeding is decreased.

The World Health Organization warns that cesarean deliveries exceeding 15% lack medical justification and may be linked to adverse maternal and child health consequences. Late initiation of breastfeeding is linked in this EHDS statistical study with cesarean delivery. Gebremedhin (2014) [34] analyzed the cesarean delivery rates in Addis Ababa Ethiopia and reported an

annual projected rate to increase 1.6% yearly in-country. Considering the impact on lactation, this increase in cesarean delivery can be seen as having an alarming impact on the early initiation of breastfeeding. The trickle-down effect of increasing cesarean deliveries can adversely impact neonatal health when there is a delay in breastfeeding initiation. The increased percentage of late initiation of breastfeeding can be viewed as a complication of cesarean deliveries when these cesareans are elective or do not provide clear justification for the cesarean delivery [34–37].

Unplanned pregnancy [**AOR = 1.35(1.1–1.65)**] has significant effect on the initiation of breastfeeding. Research has supported this finding globally that unplanned pregnancy has a significant impact on breastfeeding initiation, as documented in Ethiopia, Iran, Canada and the United States, [3, 26, 27, 38]. Factors impeding breastfeeding according to the literature include psychological stress related to an unplanned pregnancy. Women who have postpartum adjustment disorders or depression related to unwanted pregnancies may also be more inclined to choose bottlefeeding their newborn and are not interested to have baby as result interrupt the skin to infant bonding leads to ineffective early initiation of breastfeeding. The role of the health care worker in antenatal education and postpartum support to encourage breastfeeding all babies whether a pregnancy is mistimed or unwanted may be proactive in improving early onset of lactation.

Maternal age under group of 20–34 years old [AOR = 0.77(0.61–0.98)] were 77% times more likely to have early initiation of breastfeeding as compared to mothers who were youngerthan 19 years of age. This study finding is supported by the study done in Addis Ababa, Ethiopia [19]. With regard to maternal age, mothers under the age of 20–34 are more likely to initiate breastfeeding within one hour in our population. Even though the effect of older age on breastfeeding initiation remains to be elucidated [19], it has to be taken into account that an increased maternal age at first childbirth has been recorded in most high income countries in the past decades of years [39].

Despite Cesarean Section is a life-saving procedure for both the mother and the baby. Premature and wrong decision may increase the maternal and fetal morbidity and mortality and delaying initiation of breastfeeding timely. Therefore educing cesarean deliveries as a way to increase early onset of lactation, then several interventions need to target both the education of professionals and of the public. Pain control during labor and delivery needs to be addressed, in addition to maternal preference, especially among the educated, those in private hospitals, and those choosing elective cesarean deliveries [34, 36].

In connection to this, advancing maternal age has consistently been reported to be associated with a higher cesarean section rate, due to delayed childbearing is a result of increasing numbers of late and second marriages, women's growing concentration on their careers, and advanced assisted reproductive technologies, as well as increasing financial concerns that create disincentives for raising children; as a result early initiation of breastfeeding success is strongly affected [37].

## Limitation

This study was using huge national data to analyze and interpret the findings; however, temporal cause and effect relationship might not be possible due to the nature effect of cross sectional study design.

## Conclusion

Cesarean delivery adversely affects the initiation of breast feeding. Cesarean delivery, unplanned pregnancy, and being primiparous, were associated with late initiation of breastfeeding.

Women between the ages of 20–34 years of age had a reduced chance of late initiation of breast-feeding. Providing counseling regarding the strategy and importance of early initiation of breast feeding, could have crucial importance for the mother and her newborn. As early initiation of breastfeeding is related to successful exclusive breastfeeding for 6 months, this research is an important step to increasing Ethiopian practices related to lactation for maternal and child health.

## Acknowledgments

We are indebted to the DHS Program for providing us permission to use the 2016 EDHS data for this analysis.

## Author Contributions

**Conceptualization:** Getnet Gedefaw, Asmamaw Demis.

**Formal analysis:** Getnet Gedefaw, Martha H. Goedert, Eskeziaw Abebe.

**Investigation:** Getnet Gedefaw.

**Methodology:** Getnet Gedefaw, Eskeziaw Abebe, Asmamaw Demis.

**Software:** Getnet Gedefaw.

**Validation:** Getnet Gedefaw, Asmamaw Demis.

**Visualization:** Getnet Gedefaw.

**Writing – original draft:** Getnet Gedefaw, Martha H. Goedert, Eskeziaw Abebe, Asmamaw Demis.

**Writing – review & editing:** Getnet Gedefaw, Martha H. Goedert, Eskeziaw Abebe, Asmamaw Demis.

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
