## [Decision Letter · Decision Letter 0]

27 Oct 2020

PONE-D-20-25202

Effect of cesarean section on initiation of breast feeding:  Findings from 2016 Ethiopian Demographic and Health Survey

PLOS ONE

Dear Dr. Gedefaw,

Thank you for submitting your manuscript to PLOS ONE. After careful consideration, we feel that it has merit but does not fully meet PLOS ONE’s publication criteria as it currently stands. Therefore, we invite you to submit a revised version of the manuscript that addresses the points raised during the review process.

We look forward to receiving your revised manuscript.

Kind regards,

Hein Min Tun, B.V.Sc., M.Sc., Ph.D.

Academic Editor

PLOS ONE

Journal Requirements:

2. For more information on PLOS ONE's expectations for statistical reporting, please see https://journals.plos.org/plosone/s/submission-guidelines.#loc-statistical-reporting. Please update your Methods and Results sections accordingly.

Reviewers' comments:

Reviewer's Responses to Questions

**Comments to the Author**

1. Is the manuscript technically sound, and do the data support the conclusions?

Reviewer #1: Yes

2. Has the statistical analysis been performed appropriately and rigorously? 

Reviewer #1: Yes

3. Have the authors made all data underlying the findings in their manuscript fully available?

Reviewer #1: No

4. Is the manuscript presented in an intelligible fashion and written in standard English?

Reviewer #1: Yes

5. Review Comments to the Author

Reviewer #1: Comments to the Author

In the current study, the authors investigated the relationship between caesarean section and breastfeeding initiation of 7115 participants from the 2016 Ethiopian Demographic Health Survey (EDHS). The authors found that caesarean section was associated with late initiation of breastfeeding. The work is interesting and important for maternal and child health in Ethiopian, and I have a positive view of the manuscript.

However, I have some comments that I would like to share with the authors and in my view, they can help to improve the manuscript.

Background section

1. The authors stated research supports that delayed breastfeeding initiation was associated with caesarean delivery, but it lacked of citing higher level evidence to support this given that there are some well conducted systematic reviews in this field recently (see10.3945/ajcn.111.030254 and 10.1007/s10995-017-2369-x).

2. The authors need to clarify the public health implementation of investigating the relationship between caesarean delivery and early breastfeeding initiation, particularly in the setting of Ethiopia, and need to clarify the potential difference between higher income countries and low-middle income countries.

Methods section

3. Please cite the source of definition of initiation of breastfeeding. Is it from WHO or UNICEF definition?

4. Please clarify what are covariates and what are confounders as these are two different conceptions in association analysis.

5. Please use univariable logistic regression and multivariable logistic regression throughout the manuscript.

6. The authors used a p-value of less than 0.25 as a cut-off to select candidate variables for further multivariable regression analysis, but this lack of reference to support this procedure. Thus, please cite relevant statistical analysis reference after this statement.

Results section

7. It’s a bit confusing in the result part that the main research question to be answered should be the relationship between caesarean section and breastfeeding initiation other than the associated factors of breastfeeding initiation. Please focus on the research question and discuss potential roles of confounders and covariates in Discussion part.

Discussion section

8. The prevalence of delayed or late initiation of breastfeeding among women who delivered by caesarean section reported in the second paragraph of Discussion section was wrong, although this figure should have been reported in Results section. It actually should be calculated as 101/(101+64) *100% if figures in Table 3 were correct. Please double check similar wrong calculations throughout the manuscript.

9. Please clarify why associated factors of breastfeeding initiation (i.e., primipara women, unplanned pregnancy and maternal age) were discussed in the manuscript. Instead, the roles of these factors on the pathway between caesarean section and breastfeeding initiation should be thoroughly discussed.

6. PLOS authors have the option to publish the peer review history of their article (what does this mean?). If published, this will include your full peer review and any attached files.

Reviewer #1: No

---

## [Author Response · Author response to Decision Letter 0]

5 Nov 2020

Manuscript Number: PONE-D-20-25202

Effect of cesarean section on initiation of breast feeding: Findings from 2016 Ethiopian Demographic and Health Survey

Dear Academic Editor 

PLOS ONE

We thank ‘PLOS ONE’ for giving us the opportunity to resubmit this manuscript and we thank the reviewers for their constructive comments and feedbacks. We confirm that we have read the instructions for the authors and respond below to the comments on point-by-point basis. Changes are shown in track changes in the text. We hope you will find our responses satisfactory, and hope that you will find this manuscript acceptable for publication in your journal.

On behalf of all the authors

Getnet Gedefaw

Woldia University, College of Health Sciences, Department of Midwifery

gedefawget@gmail.com

Response for Reviewers Comment (Reviewer #1)

In the current study, the authors investigated the relationship between caesarean section and breastfeeding initiation of 7115 participants from the 2016 Ethiopian Demographic Health Survey (EDHS). The authors found that caesarean section was associated with late initiation of breastfeeding. The work is interesting and important for maternal and child health in Ethiopian, and I have a positive view of the manuscript. However, I have some comments that I would like to share with the authors and in my view, they can help to improve the manuscript.

Thank you very much for your appreciation for our work.

1. The authors stated research supports that delayed breastfeeding initiation was associated with caesarean delivery, but it lacked of citing higher level evidence to support this given that there are some well conducted systematic reviews in this field recently (see10.3945/ajcn.111.030254 and 10.1007/s10995-017-2369-x).

Response: Your feedback is appreciated. We read them carefully and updated our background using these recent evidences.

2. The authors need to clarify the public health implementation of investigating the relationship between caesarean delivery and early breastfeeding initiation, particularly in the setting of Ethiopia, and need to clarify the potential difference between higher income countries and low-middle income countries.

Response: thank you very much for your suggestion. We amended according to your comments under the section of introduction

3. Please cite the source of definition of initiation of breastfeeding. Is it from WHO or UNICEF definition?

Response: Thank you very much. We have cited it and taken from WHO

4. Please clarify what are covariates and what are confounders as these are two different conceptions in association analysis. 

Response: Thank you very much. Covariates are independent variables that may predict the outcome of interest and covariates may not be confounders. Furthermore, to control confounders we used multiple variable logistic regressions during the analysis phase.

5. Please use univariable logistic regression and multivariable logistic regression throughout the manuscript.

Response: Thank you very much. We strongly accepted your suggestion

6. The authors used a p-value of less than 0.25 as a cut-off to select candidate variables for further multivariable regression analysis, but this lack of reference to support this procedure. Thus, please cite relevant statistical analysis reference after this statement

Response: Thank you. We cited and put the reference 

7. It’s a bit confusing in the result part that the main research question to be answered should be the relationship between caesarean section and breastfeeding initiation other than the associated factors of breastfeeding initiation. Please focus on the research question and discuss potential roles of confounders and covariates in Discussion part.

Response: Your feedback is appreciated. We amended it accordingly. We move down to the discussion section of the manuscript.

8. The prevalence of delayed or late initiation of breastfeeding among women who delivered by caesarean section reported in the second paragraph of Discussion section was wrong, although this figure should have been reported in Results section. It actually should be calculated as 101/(101+64) *100% if figures in Table 3 were correct. Please double check similar wrong calculations throughout the manuscript

Response: Exactly you are correct. We made error while we calculate in cross tab.

9. Please clarify why associated factors of breastfeeding initiation (i.e., primipara women, unplanned pregnancy and maternal age) were discussed in the manuscript. Instead, the roles of these factors on the pathway between caesarean section and breastfeeding initiation should be thoroughly discussed.

Response: Thank you very much. We have accepted and modified it.

---

## [Decision Letter · Decision Letter 1]

7 Dec 2020

Effect of cesarean section on initiation of breast feeding :   Findings from 2016 Ethiopian Demographic and Health Survey

PONE-D-20-25202R1

Dear Dr. Gedefaw,

We’re pleased to inform you that your manuscript has been judged scientifically suitable for publication and will be formally accepted for publication once it meets all outstanding technical requirements.

Kind regards,

Hein Min Tun, B.V.Sc., M.Sc., Ph.D.

Academic Editor

PLOS ONE

Additional Editor Comments (optional):

Reviewers' comments:

Reviewer's Responses to Questions

**Comments to the Author**

1. If the authors have adequately addressed your comments raised in a previous round of review and you feel that this manuscript is now acceptable for publication, you may indicate that here to bypass the “Comments to the Author” section, enter your conflict of interest statement in the “Confidential to Editor” section, and submit your "Accept" recommendation.

Reviewer #1: All comments have been addressed

2. Is the manuscript technically sound, and do the data support the conclusions?

Reviewer #1: Yes

3. Has the statistical analysis been performed appropriately and rigorously? 

Reviewer #1: Yes

4. Have the authors made all data underlying the findings in their manuscript fully available?

Reviewer #1: No

5. Is the manuscript presented in an intelligible fashion and written in standard English?

Reviewer #1: Yes

6. Review Comments to the Author

Reviewer #1: (No Response)

7. PLOS authors have the option to publish the peer review history of their article (what does this mean?). If published, this will include your full peer review and any attached files.

Reviewer #1: No

---

## [Editor Report · Acceptance letter]

9 Dec 2020

PONE-D-20-25202R1 

Effect ofcesarean section on initiation of breast feeding: Findings from 2016 Ethiopian Demographic and Health Survey 

Dear Dr. Gedefaw:

I'm pleased to inform you that your manuscript has been deemed suitable for publication in PLOS ONE. Congratulations! Your manuscript is now with our production department. 

Kind regards, 

on behalf of

Dr. Hein Min Tun 

Academic Editor

PLOS ONE